# Empirical Myoelectric Feature Extraction and Pattern Recognition in Hemiplegic Distal Movement Decoding

**DOI:** 10.3390/bioengineering10070866

**Published:** 2023-07-21

**Authors:** Alexey Anastasiev, Hideki Kadone, Aiki Marushima, Hiroki Watanabe, Alexander Zaboronok, Shinya Watanabe, Akira Matsumura, Kenji Suzuki, Yuji Matsumaru, Eiichi Ishikawa

**Affiliations:** 1Department of Neurosurgery, Graduate School of Comprehensive Human Sciences, University of Tsukuba, 1-1-1 Tennodai, Tsukuba 305-8575, Ibaraki, Japan; s1936046@s.tsukuba.ac.jp; 2Center for Cybernics Research, Institute of Medicine, University of Tsukuba, 1-1-1 Tennodai, Tsukuba 305-8575, Ibaraki, Japan; 3Department of Neurosurgery, Institute of Medicine, University of Tsukuba, Tennodai 1-1-1, Tsukuba 305-8575, Ibaraki, Japan; aiki.marushima@md.tsukuba.ac.jp (A.M.); watanabe.hiroki.gb@u.tsukuba.ac.jp (H.W.); a.zaboronok@md.tsukuba.ac.jp (A.Z.); shinya-watanabey@md.tsukuba.ac.jp (S.W.); yujimatsumaru@md.tsukuba.ac.jp (Y.M.); e-ishikawa@md.tsukuba.ac.jp (E.I.); 4Ibaraki Prefectural University of Health Sciences, 4669-2 Amicho, Inashiki 300-0394, Ibaraki, Japan; matsumura.akira.ft@alumni.tsukuba.ac.jp; 5Center for Cybernics Research, Artificial Intelligence Laboratory, Faculty of Engineering Information and Systems, University of Tsukuba, 1-1-1 Tennodai, Tsukuba 305-8573, Ibaraki, Japan; kenji@ieee.org

**Keywords:** feature selection, pattern recognition, paresis, motor impairment, hand rehabilitation, upper extremity, activities of daily living, electromyography, wearable device, stroke

## Abstract

In myoelectrical pattern recognition (PR), the feature extraction methods for stroke-oriented applications are challenging and remain discordant due to a lack of hemiplegic data and limited knowledge of skeletomuscular function. Additionally, technical and clinical barriers create the need for robust, subject-independent feature generation while using supervised learning (SL). To the best of our knowledge, we are the first study to investigate the brute-force analysis of individual and combinational feature vectors for acute stroke gesture recognition using surface electromyography (EMG) of 19 patients. Moreover, post-brute-force singular vectors were concatenated via a Fibonacci-like spiral net ranking as a novel, broadly applicable concept for feature selection. This semi-brute-force navigated amalgamation in linkage (SNAiL) of EMG features revealed an explicit classification rate performance advantage of 10–17% compared to canonical feature sets, which can drastically extend PR capabilities in biosignal processing.

## 1. Introduction

Since the late 2010s, the use of human–computer interfaces (HCI) has played a leading role in neurological applications for advanced diagnostics and rehabilitation, a substantial portion of which is relied on telemedicine and wearable electronics [1,2,3,4,5]. Among these, a significant niche is occupied by motor training of affected limbs after stroke, creating a paradigm to establish procedures ‘as early as possible’ to promote a better outcome [6,7,8]. These patient-centered methods [3,9,10] rely on assistance support that allows tailoring the difficulty of motor training via exoskeletons, manipulators, etc. The rapid development of machine learning (ML) has thus allowed more widespread implementation with regard to individual data but not yet to the general biological patterns that, in terms of signal processing, are not fully delineated [4,11,12]. Such mechanisms or visual feedback-oriented scenes for motor image projections require patient input generated from volitional muscle signals or brainwaves [10,11,12,13]. Overall, widely used, non-invasive electromyography (EMG) remains the optimal intuitive and easy-to-use interface for stroke rehabilitation in contrast to other methods [12,14].

However, within the scope of EMG, the post-stroke heterogeneous discrepancies in abnormal muscle signals and subject-dependent characteristics make pattern recognition (PR) challenging or even impossible [15,16,17]. Empirically, at the stage of signal preprocessing, this distortion can be reduced but not completely eliminated by signal cleaning (rejection, decimation, and down-sampling) and filtering [3,18,19]. Further stages related to feature extraction and classification are essential for successful signal decoding; nevertheless, material limitations exist [11,19,20,21,22,23,24].

First, most supervised learning (SL) algorithms are linear and thereby limited in an approximation of classified labels during the learning [4,19,21]. Nonetheless, a supervised function support vector machine (SVM) allows avoidance of linearity (while using non-linear attributes) and is widely used in neurological signal processing [24,25]. Second, signal data obtained after acquisition should be represented by processed features since raw data are complex and degrades the algorithm’s capability in correct classification [20,21,26]. Those features have a smaller-dimensional weight and are able to disclose specific signal parameters more efficiently (compared to raw signals); however, feature selection requires domain expertise and remains challenging [2,20]. Furthermore, extracted feature properties from identical as well as different domains can cause redundancy and possible computational load (which is important for real-time use) [26,27]. Finally, in terms of practical use, the hypothetical number of feature components is limited due to model overfitting [11,26].

Over the past several decades, a multitude of novel features for myoelectrical classification were verified by SL algorithms [24,28,29,30,31,32,33,34]. However, for simple decoding tasks (e.g., forearm locomotion) in stroke-related applications or more sophisticated tasks of gesture recognition (usually a few simple hand gestures), the original time domain Hudgins’ set (TD-4) is mainly used [14,17,18,35,36,37]. To overcome explicit noise and artifact sensitivity inherent to TD components (calculations based on the signal amplitude), the frequency, fractal domain analysis, and timescale transformation (such as wavelet decomposition) are used as modifications or additions to TD-4 [21,34,37]. In this case, a failure to find optimal feature combinations creates notably varied results, having accuracy deviation rates of more than 50% even with similar feature sets [18,35,38]. Collectively, these studies (mainly conducted on chronic stroke) are connected by actual performance directly tied to the type and quantity of movement intentions [35,36,37]. To conclude, present paretic muscle feature assortments for decoding strategies resonate with research performed on healthy individuals or amputees and cannot provide complete domain expertise [33,39,40,41,42]. With this in mind, the most rational way to secure precise stroke-oriented feature selection is the validation of all possible combinations between usable feature vectors. Obviously, while the number of hand-crafted functions is limited, previous reports have outlined criteria that can be followed in selecting optimal parameters [19,20,21,22].

Essentially, deep learning (DL) solves such limitations and changes the shape of signal processing analysis from feature selection toward input data volume and deep neural network (DNN) architecture [4,26]. DL automates the feature engineering process, making those components a part of the neural network algorithm itself while, at the same time, non-linear function characteristics (obtained due to composition) establish closer mappings between input and output. Referring to the universal approximation theorem, feed-forward DNN with a finite number of neurons can approximate any continuous function [43]. In practical use, those neural networks do not require significant domain expertise and offer an advantage in decoding (compared to traditional ML) in signal processing tasks for HCI and other applications [11,14]. The only caveat is that general DNN models require a sufficient layer depth (and, thus, more neurons) to perform acceptably, resulting in more parameters to train [4,43]. Pragmatically, for clinical and technical aspects, EMG of acute stroke paresis data volumes are not achievable for DL applications in the foreseeable future [16,17]. Moreover, since PR studies in acute stroke (including the sub-acute stage) are rare, of small sample size, and incompatible due to diverse study designs, most PR for neurorehabilitation in stroke is limited to SL (without multilayer perception) [2,4,13].

Our previous study reported limited EMG feature domain expertise in paretic hand and finger movement decoding of highly variable post-acute stroke myoelectrical data, revealing that the optimal extracted feature sets in hemiplegic patients were dramatically different compared to healthy individuals [42]. Moreover, techniques for selecting features also vary, often resorting to generic algorithms, such as particle swarm optimization (PSO) or its hybrids [44,45], which are overly sensitive to size and acceleration coefficients. These are difficult to use since it is not obvious which part of the dimensional feature vector weight is essential for prediction [12,45]. In light of this, we instead evaluated a number of various domain features and their relative interaction on classification performance in order to evaluate the basic principles of feature concatenation and extraction for patient-specific conditions.

To our best knowledge, we are the first to study the brute-force combination evaluation of feature vectors obtained from EMG of acute stroke patients for gesture recognition. In order to reveal the direct dependence and compatibility of distinct features obtained from simple EMG acquisition (i.e., to avoid complexity in feature separability procedures), we initialized SVM-driven cross-validation of usable features in all possible combinations (mono-to-tetra). In this study, the SNAiL algorithm (Figure 1) was developed in order to explore the hierarchical pattern tendencies of predictors primarily based on the classification accuracy rate and feature domain via continuous semi-brute-force searches. These feature vectors were processed in a Fibonacci-like spiral net ranking system during the empirical investigation of extracted features, adjacency to the spiral-shaped mechanism for feature selection [44], and the Fibonacci ratio utilized for cross-selection between the superior feature sets [46].

The results of such a universal feature ranking allowed us to extend the comprehension of the nature of stroke-oriented EMG features, propose the optimal number of features for learning in low-channel wearables, and develop a novel, generic ranking method for myoelectrical feature selection in stroke recovery applications.

## 2. Materials and Methods

### 2.1. Participants

Given that EMG acute stroke data are rare, we re-examined a previously recorded dataset of 19 patients with 11 cases of cerebral infarction and 8 cases of intracranial hemorrhage within 1 month since onset (Table 1) [42]. Each candidate, at the moment of signal acquisition, had hemiplegia which resulted in various grades of paresis as assessed by the Stroke Impairment Assessment Set Motor (SIAS-M) instrument, only the knee–mouth test and finger-function test, the Fugl-Meyer Assessment of the upper extremity after stroke (FMA-UE), and the Brunnstrom stage of the forearm and hand scoring.

To reduce bias, participation criteria included: a registered stroke incident, age over 18 years old, and the ability to understand the observational procedure. Each candidate submitted a signed, informed agreement to participate in the study. Regarding the regulatory approval for clinical trials, the Ethics Committee and Review Board of the University of Tsukuba Hospital approved the research (R02-204) in accordance with the Declaration of Helsinki.

### 2.2. Data Acquisition and Signal Preprocessing

EMG recording during forearm and hand movements of hemiplegic patients was conducted via wireless device (8CH HUB 19022021) with a 16-bit ADC and 1000 Hz recording frequency. Four gel bipolar surface electrodes were attached to the proximal projection of major muscle bellies of forearm flexors and extensors plus an additional bipolar electrode was placed on the thenar area of the palm. During the experiment, patients performed six simple hand gestures ten times each (fist grasp, pinch, wrist flexion, wrist extension, palm opening, and thumb) while sitting in a wheelchair and leaning on an adjustable table set to accommodate the posture (Figure 2).

Filtering procedures included high-pass/low-pass frequency cutoff directly from the hardware (5 to 500 Hz), a 4th Butterworth bandpass range from 20 to 300 Hz, and a Hampel filter with every 100 neighboring samples set to the 2nd standard deviation [23,47]. Hemiplegic datasets were normalized by root mean square normalization. To specify the myoelectrical area of interest range (30 to 60 ms) within the static gestures, we implemented gesture autodetection by tracking frequency root mean square envelope and signal power peak detection within a certain segment of gesture class (i.e., separate peaks within a fist or within thumbs up). Selected events were visualized to prevent artifacts and errors.

### 2.3. Universal Feature Extraction for Classification

Aiming to explore domain hidden trends and specify undisclosed prospective feature extraction methods, a list of 127 features (Appendix B) was selected for extraction in accordance with their biomedical relation towards specific motor synergies [10,19,20,21,22,24,26,27,28,29,30,31,32,33,34,39,40,41,42,48,49,50,51,52,53,54,55,56]. Moreover, logarithmic interpretations of certain features were also used since, in the scope of abnormal myoelectrical events, particular voluminous values of extracted features (mainly for linear classifiers) can spot non-typical changes over the distorted signals and thereby increase predictive capabilities [19,32,48]. All features were concatenated into the feature vector.

### 2.4. Brute-Force Feature Selection Search

This feature vector was restructured into singular composite functions, where each particle vector (extracted from the single feature) was processed in mono-, binary-, triple-, and tetra-brute-force combinational SL with one, two, three, and four features (all possible combinations with no repeats and order) in 10-fold cross-validation (SVM with 100 iterations for each possible permutation). During the cross-validation process, the myoelectrical signatures of two out of the 19 participants were randomly chosen as the test data, and the remaining 17 participants’ myoelectrical signatures were used as the training data. This procedure was repeated 100 times. To perform 10-fold cross-validation with our 19 participants, the number of test participants was calculated as 19 × 0.1 = 1.9, and by the best approximation, data from 2 participants were used as the test data. The following mathematical Equation (1) explains the number of combinations in the brute-force search (BFS), where *n* is the number of features, *r* is the level of concatenations in between possible singular feature vectors (from 1 to 4):(1)$$BFS=\frac{n!}{r!\left(n-r\right)!}$$

For better navigation, most of all features have a short description and mathematical definitions structured by the initial domain group as follows (Appendix A): time domain (TD), frequency domain (FD), time–frequency domain (TFD), fractal domain (FRD), or spatial domain (SD) functions. These refer to the parametrical components, such as the power signal ratio, between the selected channels in different muscle groups (i.e., flexors versus extensors ratio).

### 2.5. Semi-Brute-Force Navigated Amalgamation

The obtained numerical BFS combinations (BFS1 had 127 x, BFS2 had 8001, BFS3 had 333,375, and BFS4 had 10,334,625 extracted feature set permutations) were ranked and processed into a Fibonacci-like, spiral net, post-brute-force concatenation. We refer to this as mono-to-tetra brute-force sourcing. 

The hierarchical ranking is based on a cutoff of the near-to-top quintile in the feature or multiple permutations set listing where the highest level of classification rate prevailed. Furthermore, the cross features in those areas underwent continuous numerically increasing permutations among themselves with a magnification factor of 1.618 or the Fibonacci ratio. Referring to other previously reported, differing methods [35,45], our novel feature amalgamation method includes simple, pair-wise ranking elements that implicate cross-validation between subsets to elaborate novel, semi-brute-force navigated amalgamation in linkage (SNAiL) of EMG features up to 20 feature-length generic subsets (from 5 to 20 features in concatenated vectors) in order to trace the consistency in the results. For further semi-brute-force concatenations, in contrast to previous studies that have employed aggregation-based feature selection methods from a large set of candidate features (see Appendix B), our spiral semi-brute-force search (SBFS) method generates combinations of extracted features for subsequent SL and ranking based on superior performance rates only (see Figure 3).

### 2.6. Statistical Metrics and Evaluation

In order to compare the classification performance of feature sets generated by BFS and SBFS, commonly used canonical feature sets were extracted from the same pre-processed EMG dataset. Moreover, since most of the extracted feature sets are TD and usually have a limited number of features, it was decided to select feature sets with FD, TFD, and FRD attributes from related studies. Those multiple feature sets abbreviated as MFS are as follows:Hudgin’s feature set or MFS1: MAV, WL, ZC, and SSC [28];TD-AR feature set or MFS2: RMS, and AR6 [29];TD-NLS feature set or MFS3: LMAV, and NSV [30];Du’s set or MFS4: IEMG, VAR, WL, ZC, SSC, and WAMP [31];NTDFS time domain set or MFS5: SSI, RSD1, RSD2, MSR, ASM, and ROG [32];TD-DFA feature set or MFS6: ZC, SSC, AR4, PSR, DFA, and HFD [33];Oskaei’s and Hu’s feature set or MFS7: MAV, RMS, WL, VAR, ZC, SSC, WAMP, MMAV1, MMAV2, PSP, AR2, AR5, MDF and MNF [24];Wang’s feature set or MFS8: MAV, VAR, AR-4, ZC, MNF, MDF [34];

Prior to statistical testing, several procedures were performed. For classification, patient data were normalized and randomly distributed between 17 testing and 2 training patients using SVM 10-fold cross-validation. For stabilizing and managing the learning sub-datasets obtained from randomness due to various stages of motor impairments, each calculation was performed with 100 iterations (optimal) in pre-calibration settings.

To determine whether the best empirical permutations have a higher classification rate (Equation (2)) than the other canonical feature sets, a one-tailed unpaired *t*-test was utilized (*p* > 0.001). The exaggerated simplicity of the statistical validation is due to a single dataset of 19 patients (a small sample size of the rare acute stroke data) and the study purpose (to precisely find the most effective predictive parameters obtained from the feature sets).

Moreover, 10-fold cross-validation divides the dataset into random subsets and then obtains the final classification rate or other machine learning metrics for every random case. Hence, the mean correct classification rate (CCR) obtained from the 100 iterations (which is the sample size within the statistical testing) of each cross-validation and its standard deviation (as a measure of a dispersion of a sample of data around its mean) was used for the testing.
(2)$$CCR\;\left(\%\right)=\frac{Number\;of\;correctly\;classified\;instances}{Total\;number\;of\;instances}\cdot 100$$

These experiments are focused on the classification rate performances of certain feature sets on a single dataset to determine any significant differences between the means of the two groups. Therefore, the null hypothesis states that there are no differences in the classification rates of the two different extracted feature sets whereas the alternative hypothesis signifies that a certain feature set has a higher classification rate than the other. The unpaired *t*-test is suitable for this comparison inasmuch as it is utilized to compare the means of classification rates when the variances of the compared groups are unknown and derived from cross-validation fixed to a small sample size.

Additionally, the degree of freedom (DF) was taken into consideration because it determines the appropriate test statistic to use when comparing the mean CCR between different feature sets. With a bigger DF, the test comparison could be more precise in the estimation of the population mean (i.e., the test statistic is asymptotically normal) and mitigate the dispersed standard deviation caused by the cross-validation (i.e., the standard error of the mean is related to the standard deviation).

MATLAB R2021b (MathWorks, Natick, MA, USA) was used to perform all statistical evaluations between the MFSs and most efficient empirical combinations trained on the same dataset depending on the SNAiL settings. 

## 3. Results

### 3.1. A Novel, Generic, Patient-Centric Method for Supervised Learning Classification

Our results indicate that, for ML classification, the use of semi-brute-force analysis of pre-processed EMG signal paretic movements is competitive or even superior in myoelectrical feature extraction (Figure 4).

Initially, we explored 127 of the most utilized features (see Appendix B and Appendix A) in all possible mono-to-tetra permutations to reveal essential criteria for further feature vector concatenation and learning. Through an observational study of post-stroke patients in stroke care unit facility conditions, the SVM algorithm was trained to recognize seven classes in the prediction of six common hand gestures, with the rest referring to common activities of daily living (ADLs) [9]. 

Brute-force permutations were also able to simultaneously reveal a linear trend of the increased mean correct classification rate (CCR) depending on the number of selected EMG features and specified stroke-oriented domain expertise through the prism of best combinations (Figure 4B). Finally, the consistent patterns in Figure 5 were used to design a novel generic algorithm comprised of an amalgamation of the EMG features (semi-brute-force navigated amalgamation in linkage; SNAiL) as a generic alternative to feature selection. The ML procedures and study design settings are described in the Section 2.

### 3.2. Post-Stroke Participant Characteristics

In total, the 19 stroke survivors (from 32 to 82 years old), all within one month since cerebrovascular onset, were examined during the observational study (Table 1). Each candidate, during signal acquisition, had hemiplegia which resulted in variable grades of paresis as assessed by the physical therapist.

### 3.3. An Empirical Analysis of EMG-Oriented Features

For single feature testing (Appendix A), we found that the fractal function FR4 has the most efficient CCR in gesture prediction, reaching 63.93% (BFS1). Meanwhile, other features from different groups with non-similar equations have competitive prediction scores obtained from the SVM 10-fold cross-validation (e.g., timescale EWT-8, frequency MASP, WL, and IEMG). In a comparison between FR4 and efficient feature vectors, most results were statistically significant except for the time–frequency (TFD) features [19,22].

In binary-to-tetra testing illustrated in Figure 5C, the best feature vectors tended to form various domain linkages: PERC2 and EWT10 at 68.43% (BFS2); WL, MTW, and EWP8 at 69.93% (BFS3); and SM1, SMN, EWT6, and CRD at 71.50% (BFS4). The complete definitions of used features in universal feature extraction and references can be found in the Section 2 and Appendix B. Moreover, during the initial testing, the utilization of all 127 extracted features from the EMG data as a single concatenated vector during SVM decoding yielded only a 53.25% CCR in distal hand movements prediction.

Similarly, considering the Figure 4A performance metrics, the domain distribution of the top 20% of combinations with the higher mean CCR has been illustrated in Figure 4B which shows the near-to-exponent magnification of selected permutations. Here, there is a trend in which the best combinations tend to amalgamate between different domains, whereas superior triple-to-tetra sets from a single domain are minor. Moreover, the deeper the level of permutations, the higher the percentage of multi-domain feature vectors (e.g., nearly 30% in BFS2 and more than 80% in BFS4 “TD–Other” combinations).

In order to assess this hierarchical continuity and any linkages between studied features, we visualized the top 50 sets and their transitions across the increased BFS combinational value for all possible permutations. Here, we observed that superior features from a lower BFS are highly probably further present in the superior multi-feature combinations in the higher BFSs (e.g., IEMG in mono persists in the best binary and triple sets).

This phenomenon is partially illustrated in Figure 5, which shows the relationship with superior features from the previous brute-forcing (in terms of CCR). As a result, these hidden myoelectrical trends in SL decoding were deployed for further EMG signal feature vector concatenation, and this observation has become a starting point for establishing the SNAiL algorithm (which concatenates the best features with each other and ranks them during semi-brute-forcing). The transition of mono-to-tetra (BFS1 to BFS4) in Figure 5C shows only the top 50 permutations due to the massive number of evaluated combinations.

### 3.4. Semi-Brute-Force Search Based on Hierarchical Feature Ranking

In order to establish the navigated amalgamation of features into high-dimensional vectors based on SL cross-validation, the semi-brute-force search (SBFS) strategy was used. Here, to increment the listing of semi-brute-forcing between specific feature vectors (i.e., the number of combinations for processing), we constructed a Fibonacci spiral net ranking as a magnification factor in EMG feature linkage based on CCR ratings.

In the study, the proposed navigated amalgamation or SNAiL (Figure 1) employed a systematic approach for feature selection utilizing a range of 5 to 20 features from possible extracted sets in order to optimize performance in consistent loops until deferred model overfitting (as depicted in Figure 6A). The initial iteration, SBFS5, demonstrated a decrease of less than 1% in CCR for paretic gesture prediction compared to BFS4, whereas, over the increasing number of features, the highest CCR achieved at SBFS19, representing a near to predefined limit for SNAiL.

This approach resulted in a significant improvement in performance with a 4.5% increase in CCR compared to the initial iteration (SBFS5) and near to 4% increase in CCR compared to the best result obtained through a full brute-forcing approach (BFS4). The algorithm’s self-selected multi-domain feature sets with the best CCR are described below. Based on semi-brute-force searching, the best efficiency (75.5%) in terms of classification of extracted feature sets consists of the following 19 concatenated features: MMAV5, SSI, LSSI, RSM0, MSR, MHW, MTW, AR3, AEN, MLASP, SMN, FDD, STFT, EWP-6 and 10, EWT-4, HHT, SWT, and DWT.

In order to evaluate the practical limits and capabilities of SNAiL, we investigated the effects of utilizing various brute-force source origins (Figure 5A,B) on feature vector concatenation. This was achieved by conducting semi-brute-force permutations of mono-to-triple and mono-to-binary brute-force input data sources, as shown in Figure 6B,C. The results of 10-fold cross-validation indicated that the use of a mono-to-triple brute-force source in BFS1-3 (Figure 5b) yielded performance gains in EMG decoding comparable to those obtained using BFS1-4 (Figure 6A,B). Likewise, utilizing a mono-to-binary source (BFS1-2) produced similar results (i.e., BFS1-3 and BFS1-4), although with a higher standard deviation and lower CCR. Despite the reduced performance in Figure 6C, the results were found to be statistically significant throughout testing, having similar classification trends. Furthermore, it was noted that the use of limited brute-force input data for SNAiL resulted in a drastic reduction in computation time during spiral ranking and semi-brute-forcing. In these three scenarios on optimal vector concatenation, the best feature sets tended to be similar to the various deviations. In Figure 6B the SMBS20 contains the following 20 features: MMAV5, SSI, ASM, EWL, MHW, LCARD, MASP, MNP, FDD, STFT, EWT-4, 6, 8 and 10, EWP-6, HHT, SWT, DWT, CC-R, and FR4 for a 76.10% CCR rate. Whereas results for SMBS20 (Figure 6C) with IEMG, MMAV5, VAR, RSM0, RSD1, RSD2, WL, SSC, MHW, MTW, LCARD, AEN, MASP, SM2, EWT-4, 6 and 10, HHT, CC-R, and FR4 have a 73.15% CCR rate. Notably, these current feature sets include the features from each studied domain with the TFD elements foremost.

Excluding pre-calibration, the interval calculation time took 1800 h. The processing was performed simultaneously on a custom-made, parallel computing cluster using 18 consumer-type computers and MATLAB scripts (Appendix A). The total number of parallel MATLAB terminal processes during the BFS and SBFS tasks was 342 (simultaneous calculations).

The parallel computing units of MATLAB (for Windows OS 10 and 11 Pro) terminals for predefined segments classification included: five units with Intel^®^ Core™ i7-6700 CPU 3.40 GHz (Intel Corporation, Santa Clara, CA, USA) and 16 GB of RAM (six MATLAB processes for each of four units), one of them was used as a custom-made server in order to save temporal files on a single PC; one unit with Intel^®^ Core™ i9-7900X 3.30 GHz and 128 GB of RAM (22 processes) which was also used as a master controller to monitor and allocate tasks of each of the computers; one unit with Intel^®^ Core™ i9-9900K processor and 3.6 GHz with 32 GB of RAM (14 processes); four units with 12th Gen Inter^®^ Core™ i5-12400 2.5 GHz and 64 GB of RAM (25 processes for each); two units with 11th Gen Intel^®^ Core™ i9-11900K 3.50 GHz and 64 GB of RAM (26 processes for each); two units of 12th Gen Intel^®^ Core™ i7-12700 2.1 GHz and 32 GB of RAM storage (28 processes for each); two units with AMD Ryzen^®^ 5 Pro 5650 GE (Advanced Micro Devices Inc., Santa Clara, CA, USA) and 64 GB of RAM (16 processes for each); and one unit with AMD Ryzen^®^ 9 5900X 12-core processor at 3.70 GHz and 64 GB of RAM (42 processes). 

For each PC, processor overclocking was not performed due to overheating and management concerns. Both SSD and HDD types of storage were used during the aggregation of massive EMG data.

### 3.5. Statistical Comparison across the Extracted Feature Sets

For comparison, the most superior feature sets obtained from the SBFS and BFS were tested versus MFSs (Figure 7). Statistical significance during the unpaired *t*-testing indicated that the feature sets obtained from SNAiL are significantly superior in hemiplegic gesture prediction (Table 2). As a rule, the higher *p*-values were obtained while comparing the best BFS and SBFS feature sets.

For obtaining the CCR, we used 10-fold cross-validation with 100 iterations each (in which, for every comparison, n equals 100). Here, the obtained degree of freedom (DF) is the same because it references the number of learning iterations of 198 independent observations used in the comparison. These results suggest that scores in Figure 7 have enough power to detect meaningful differences in CCR between the different feature sets.

## 4. Discussion

We found, using a low-channel system, that our novel approach in feature selection has a highly significant (*p* < 0.001) performance increase of 10–17% in gesture prediction compared to the state-of-art feature extraction methods as shown in Figure 7.

Here, results obtained from parallel cluster calculations (over 1800 h) illustrate that a novel, generic approach is capable of higher performance in hand gesture EMG prediction, amidst post-stroke hemiplegic patients, in spite of relatively high dispersion during the 10-fold cross-validation (i.e., based on paresis grade and inconsistency of different folds). The results suggest that the SNAiL approach offers a flexible alternative for interpreting bioelectrical signals from non-healthy people [17,18,33]. On the whole, the algorithm addresses challenges that rise when canonical feature extraction methods can lead to inconsistent performance or poor classification capabilities [35,36]. It achieves this by using a post-brute-force hierarchical coupling process, which combines multiple feature vectors into a single valid input for the classifiers (see Figure 3 and Section 2.5). This method also helps to reduce the amount of data required for training. However, several aspects of our research design and SNAiL algorithm should be specified in more detail.

By introducing empirical methods in EMG feature domain expertise [20,57], our study aimed to enhance the potential of hemiplegic movement decoding [17], minimize the importance of feature selection procedures, parse dimensionality and complexity of extracted feature vectors, and verify relationships with a number of selected channels (the volume of data for the training) [4,37]. In other words, our method extends the comprehension of paretic muscle behavior and provides exact feature domain expertise in PR.

In addition to normal challenges in upper extremity EMG data gathering using surface electrodes (e.g., muscle signal crosstalk, spasticity, and local patient conditions), stroke presents additional challenges since post-stroke muscles tend to develop not only functional but also anatomical changes [15,38]. All of the above affect both the quality of the EMG recordings and extracted features in terms of signal decoding. For instance, in a study comparing upper extremity movements between hemiplegic and healthy individuals [58], it has been observed that MNP and MNF (Appendix B) FD features exhibit chaotic behavior in the dynamic signal changes during contractions. Moreover, these features may not display similar signal trends as healthy movements. Such an analogous phenomenon was observed in our previous study on acute stroke patients [42].

Referring to practical use for neurological patients, the use of EMG wearables with limited computational capabilities poses challenges for the implementation of real-time HCI in rehabilitation [2,13,45]. Thus, the scope of ML is trending towards searches of optimal feature sets, consisting of a reduced number of features, as well as signal transformations or other methods to enhance the meaningful feature vector weights [10,26,41]. In addition, it has been revealed that even identical EMG features could behave differently in prediction tasks based on the signal frequency which may vary by device [23]. Therefore, we wanted to focus on mono-to-tetra brute-force-discovered phenomena with minimal extraction from the signal feature vectors (Figure 5A–C).

In the initial brute-forcing, the feature domain expertise turns out to be relatively unimportant. For generation BFS1 with a single vector, only several out of eighty TD features (IEMG, SSI, RSM0, NSV, WL, and MHW) were sufficient for the group’s highest classification rates of around 60–62% (Appendix A), thereby being inferior versus the fractal dimension FR4 feature. Although these TD features indicate different aspects of muscle activity, they are complementary to each other in terms of signal information processing [30,32,39,49]. This and the obtained results could indicate that IEMG and SSI or WL and NSV feature vector extraction from the raw paretic EMG are oriented towards related or similar events in paretic muscle behavior during function approximation. At the same time, results identical to the TD performance were obtained using TFD (EWT-8, DWT, and HHT) and even FD features, for which such domains are rarely found in the most efficient results while using the mono-feature extraction method [20]. Derived from the signal’s power spectrum density, the limitations of the FD features can be associated with the fact that muscle activity may not be periodic, especially in cases of irregular and complex signal patterns of paretic contractions [59]. For MASP, the observed classification can be explained by the feature’s sensitivity to spasticity or muscle weakness by measuring the changes in amplitude of the dominant frequency peak (or its absence), making it a utility feature in abnormal EMG PR [60]. As a rule, these above-described features from the mono ranking were found in the best feature set combinations generated by SNAiL or navigated amalgamation of EMG features using semi-brute-forcing.

We also studied features with a non-linear relationship between muscle force and EMG amplitude [30], since state-of-the-art studies suggest that non-scale descriptors are useful in gesture classification [32,48]. In our study, non-scaled features did not result in improved performance when used alone in the raw paretic EMG dataset, except for NSV and ER hand-crafted variations. However, non-scale descriptors LSSI and MLASP were found among the most efficient feature sets while using SBFS.

As shown in Figure 4B, in binary-to-tetra permutations (BFS2-4), the analysis of the most efficient 20% permutations revealed certain tendencies in which the leading mean CCR in movement attempts prediction was obtained if the extracted sets had more features in sets and different domains were combined. Likewise, identical observations using linear SL classifiers were yielded across the myoelectrical decoding studies where time and frequency domains were combined into a single input [26,45]. However, when features are not relevant or contain similar information about the signal, it may cause classifier overfitting or feature redundancy that decreases biosignal processing [11]. To a certain degree, this trade-off can be revealed with feature ranking or by different SL algorithms [4,19,42].

Furthermore, with the advantage of full brute-force analysis, we revealed that most efficient feature sets are similar in terms of components (e.g., triple-vector WL-MHW-EWT6 and WL-MTW-EWT4 at BFS3) and translate hierarchical carriages of TD or other domain features from mono-to-tetra as depicted in Figure 5. For instance, the MHW feature from BFS1 amalgamated with SSI at BFS2, then EMG feature linkage transformed into the MHW-SSI-FR4 feature vector at BFS3 before finally concatenating into MHW-SSI-FR4-EWT8 at BSF4. This conclusion is supported by the consistent performance observed during 10-fold cross-validation.

For BFS4 with more than ten million empirical variations, the distribution of top features shown in Figure 4B disclosed unique combinations that did not appear in mono or binary ranking in accordance with the hierarchical ranking transition. For illustration, the features set composed of SM1-SMN-EWT6-CRD (which lacks TD attributes that do not consider the spatial MAV ratio between the electrodes) and MMAV5-MTW-AEN-MNPD (without the TFD features) were found in the top ranking. In comparison to different feature extraction methods, the best BFS4 turns out to be significantly better in decoding EMG paretic data. Out of the MFSs shown in Table 2, a higher CCR was obtained from canonical Hudgin’s TD-4, Du’s, and TD-NLS feature sets [28,30,31]. This might support the statement that the feature quality, but not the quantity, determines the linear SL potential in myoelectrical PR [21]. As the above suggests, it is not the domain describing the signal, but the unique combination that plays the main role in efficient feature concatenation and, by extension, classification.

We believe that the progressively growing composition of the feature sets through our SNAiL algorithm emulates a spiral evolution. Our SNAiL algorithm incorporates a proportional exponent of the Fibonacci number (i.e., the golden ratio) when truncating the ranked feature sets for each SBFS iteration. As a result, the step-by-step composition of the ranked feature sets reflects the evolution observed in the Fibonacci sequence. Mathematically, the Fibonacci sequence corresponds to the formation of a spiral shape in geometric representations. This type of Fibonacci-based spiral evolution is widely observed in nature, exemplified by the morphological characteristics of various organisms, including sunflowers, pinecones, elephant tusks, seahorse tails, and many others including the shell of a snail [61,62].

During the SNAiL spiral concatenation of features, the Fibonacci ratio proved crucial in the generation of candidate vectors for extraction from the signal and SVM cross-validation. Specifically, the first spiral ranking in SBFS4 obtained nearly the exact same (or had three out of four elements) combinations of features from the full brute-forcing using moto-to-tetra input, at the same being statistically outstanding in gesture recognition in the most canonical feature sets (*p* < 0.001) with a 10–12% classification increase gap (Appendix A). The parallel computing calculations of BFS4 (more than ten million combinations with one hundred iterations for each) took more than a month while SBFS4 (see Figure 6B) was performed in minutes. This hierarchical feature ranking avoids the unnecessary empirical processing involved in BFS. Finally, it shows that the limited brute-force input for model classification is still able to generate sufficient increase scores compared to BFS and is significantly better versus other non-empirical feature extraction methods by 10–17% (Figure 7). Thus, SBFS using binary or triple brute-force sourcing outperforms traditional feature sets and even computationally intensive ‘snail-paced’ BFS for feature selection in EMG PR. This approach reinforces the importance of patient-specific signal preprocessing for more accurate ML models [33,35]. Different populations, with differing degrees of motor impairment, may have various factors that affect motor control and produce notably unique stochastic EMG signatures [2,12]. Therefore, patient-specific calibration is crucial in successful prediction and should be prioritized.

In a spiral concatenation of vectors using SNAiL having full or fragmental brute-sourcing feature listings, amalgamation in between leads to similarly optimal feature sets for SVM decoding. As seen in Figure 6, for every method, the best permutations have more than half of their features as identical, indicating that features tend to organize independently from specific settings.

These findings are consistent with previous studies that have used TFD features for the decoding of chronic stroke movements [37]. Thereby, even with an increased number of features (i.e., SMBS20), it is clear that SNAiL appears to reduce overfitting by identifying and isolating near-identical features or similar modified components (i.e., SSI or non-scale LSSI). In particular, we have identified features that, after brute-force searching and semi-brute-force generation, preserved unique qualities in the correct classification of recognizing multiple gestures in a multi-class SVM classification: MMAV5, RSM0, SSI, MASP, HHT, and DWT. This reinforces the position that different features are able to emphasize different extracted features even while being combined as a single vector for a more optimal approximation, therefore generating a better correct classification or any other metric. The results from the higher-layer feature permutations suggest that, while the MHW (and MTW) feature has been previously shown to be important in muscle fatigue and gesture recognition [20,49], other features also play a significant role in improving the classification accuracy of SL approximation.

We acknowledge the limitations of our study. First of all, the SNAiL deployment requires a certain threshold in domain expertise to utilize EMG features. However, this concern is the same across feature selection techniques using linear SL [25]. We also observed an inconsistent variability of performance metrics caused by the feature splitting. This could indicate that unique combinations that took place in BFS4 did not appear during the semi-brute-force between only efficient features. Finally, it is important to note that the used mean CCR is only an approximation of the model’s performance and may not accurately generalize to EMG signal data outside the training dataset. In general, machine learning metrics, such as F1-scores or precision, could provide a more robust evaluation [30,33]. However, in the context of our empirical study, which compared a proposed feature generation method versus different feature extraction methods from a limited population of acute stroke patients using 10-fold cross-validation, the use of CCR does not detract from the key findings that the SNAiL approach leads to a significant improvement in myoelectrical PR. Lastly, we also recognize the time-consuming process during the BFS process as a limitation that required a certain threshold for practical utilization.

In future work, we plan to evaluate the SNAiL approach using alternative classifiers such as k-nearest neighbors (k-NN) or random forest trees (RFT) [39]. Additionally, we intend to improve prediction capabilities by incorporating random feature insertion using genetic algorithms during the semi-brute-force search process to prevent the loss of useful features [44,45].

In terms of stroke rehabilitation, this method can be applied to most EMG-driven biofeedback therapies such as lower-limb post-stroke training or facial paresis recovery. This is important because even short additional exercises can significantly increase rehabilitation outcomes [7,8]. However, the general principles of the proposed SNAiL algorithm and the variety of selected PR features used during its validation (such as Hjorth complexity [50]) allow for extrapolating the deployment of this spiral algorithm in different biosignal processing applications such as EEG (see Figure 8).

As there is expected to be a great disparity between the number of stroke patients in the near future and the number of physicians, advanced methods for monitoring recovery and biofeedback interactions are essential and any methods reinforcing those applications would be welcome [2,3,4,5].

For this reason, our results support the SNAiL approach due to its simplicity and ability to mitigate challenges caused by large amounts of training data. Taking into consideration the proposed algorithm, the use of intricate DL networks may be redundant in most hemiplegic EMG signal processing tasks. The presented multi-domain universal feature collector (Appendix B) can be used as a starting point with the option to adjust or reduce functions as needed for specific goals or applications.

In conclusion, we discovered that binary- and triple-EMG signal, feature vector, and brute-force concatenations (for comparing up-to-date features) are sufficient input to generate superior feature sets for ML classification in hemiplegic distal movement decoding. This approach is urgent for practical usage in placing patient-specific characteristics first in the field of neurological rehabilitation after stroke or injury.

## Figures and Tables

**Figure 1 bioengineering-10-00866-f001:**
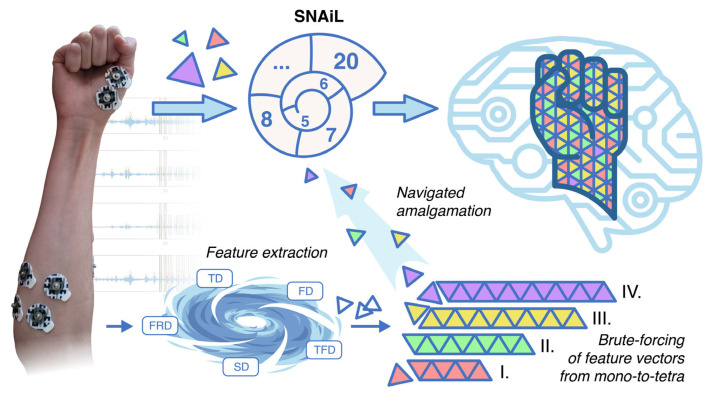
After collecting EMG in stroke patients, the data are combined into a single dataset before a number of features are extracted into a concatenated feature vector. The feature vector manifold splits data into low-dimensional particles for a brute-force coupling of mono-, binary-, triple-, and tetra-cross-validation using SL. Finally, post-brute-force vector particles are organized in a Fibonacci-like spiral ranking net and are further compressed into high-dimensional, best-accuracy feature sets for myoelectrical hand gesture recognition.

**Figure 2 bioengineering-10-00866-f002:**
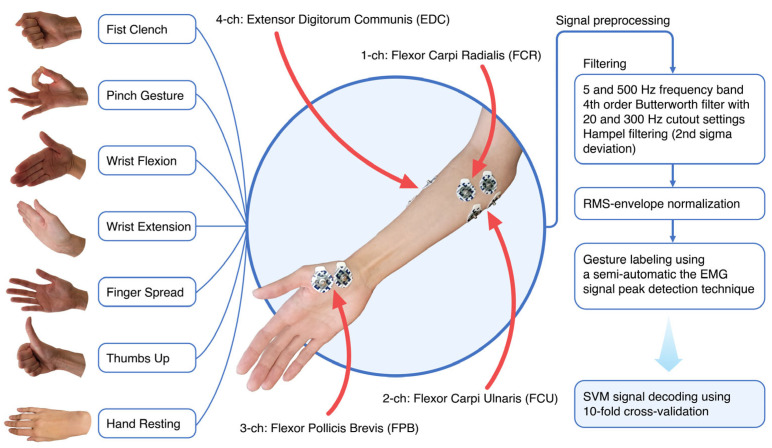
EMG signal acquisition was extracted from the forearm and thenar area of the hand. Filtering and frequency cutoff settings for pre-processed myoelectrical hand gesture signals reduce signal noise and artifacts. For the purpose of training, in every case except the rest state, the hand gesture signature locus in the filtered signal was selected using the peak detection approach (30 to 60 ms range offsets with 1/3 supremum and 2/3 infimum borders of the peak) in the single gesture sub-sector.

**Figure 3 bioengineering-10-00866-f003:**
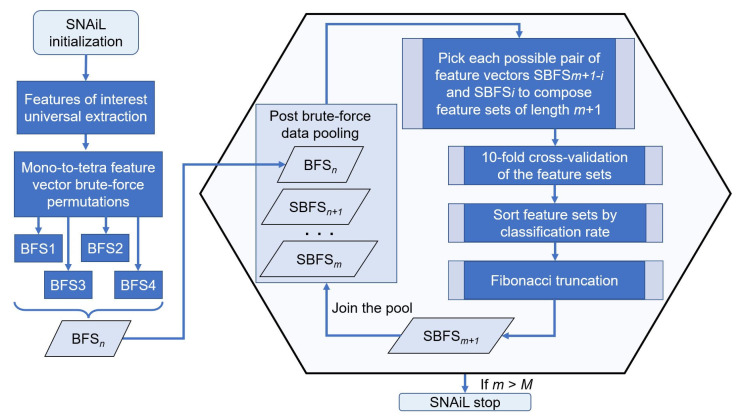
SNAiL spiral algorithm description. The algorithm initialization starts with extracting features of interest and further brute-force generation of empirical feature sets permutated from the singular components of the studied feature list. Those mono-to-tetra permutations are pooled as an input source for high-dimensional navigated amalgamation (or concatenation), aiming to obtain the most efficient feature sets. Each combination is processed by 10-fold cross-validation and further sorted by the obtained classification rates for each feature set. The captured rating ratio is truncated in accordance with the Fibonacci value (*Fn*) up to integer numerical values suitable for second-rank cross-validation. Obtained combinations of values are reimplanted into the existing data pool (with the brute-force input) and procedures are continuously repeated until set conditions are made. *n*—numerical index of brute-force layering, *m*—numerical index of semi-brute-force layering, *i*—numerical index of iteration when composing feature sets, *M*—predefined maximum limit of spiral concatenation.

**Figure 4 bioengineering-10-00866-f004:**
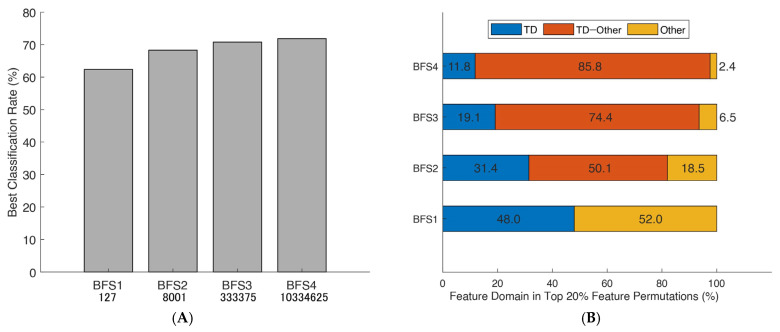
Best classification rates obtained from brute-force searches and their domain distributions: (**A**) the maximum classification rates of seven paretic hand gesture predictors obtained from brute-forcing extracted feature vectors using SVM 10-fold cross-validation in 19 patients; and (**B**) a breakdown of the feature domains of the top 20% best combinations, with the proportion of specific domains depicted in color and based on time domain features (TD). The brute-force search (BFS) considered permutations of mono-to-tetra feature vectors.

**Figure 5 bioengineering-10-00866-f005:**
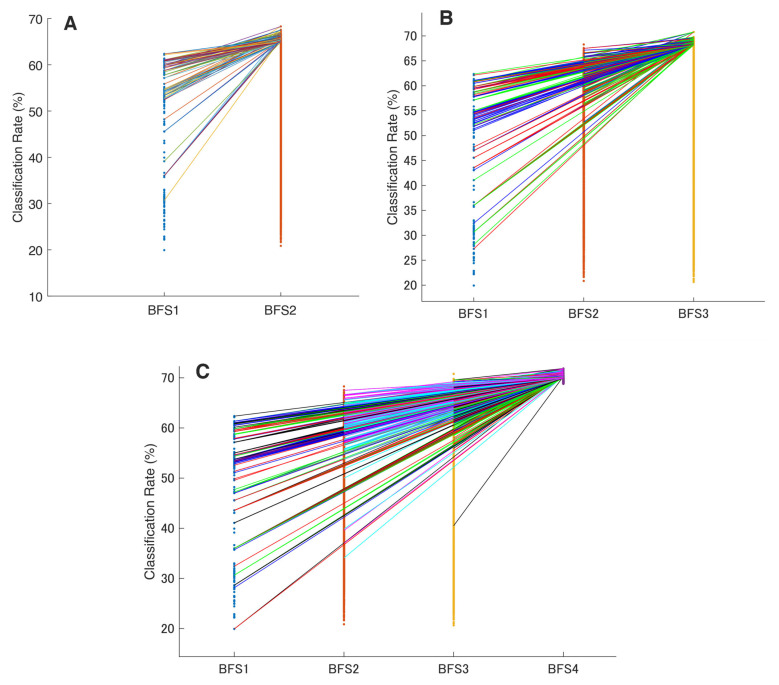
The hierarchical linkage of the extracted feature vectors in all possible brute forcings. These figures illustrate the list of the top 50 feature sets and their transition during the permutated layering in classification rate ranking: (**A**) in mono-to-binary; (**B**) in mono-to-triple; and (**C**) in the mono-to-tetra inspection. The highlighted color lines between the bars show the transition of the best feature sets and their further appearance throughout the ranking. The horizontal bars made of single-color dots show the classification rate of every permutation (except for the BFS4 where only the top 50 combinations are visualized).

**Figure 6 bioengineering-10-00866-f006:**
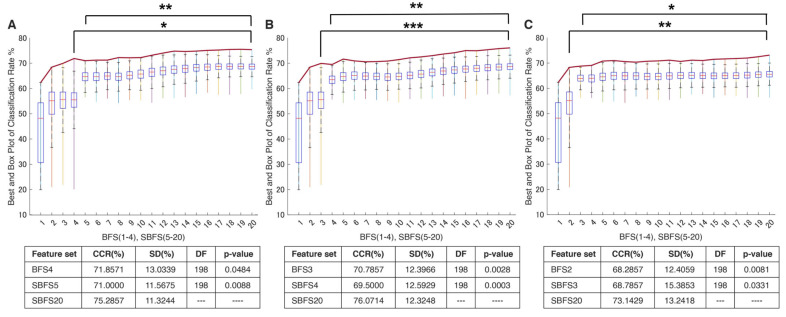
Classification performance using the SNAiL approach in multiple paretic gesture prediction using different brute-force origins. In every figure, columns show the classification rate (with the red line indicating the average classification rate) of the best brute-force (BFS) and semi-brute-force (SBFS) permutations of superior concatenated feature vectors in gesture recognition of 19 acute stroke patients. Error bars are depicted for the top quintile of combinations: (**A**) Results obtained using mono-to-tetra, (**B**) mono-to-triple, and (**C**) mono-to-binary brute-force input data. CCR—correct classification rate, SD—standard deviation, and DF—degree of freedom during the unpaired *t*-test. * shows statistical significance between the SBFS and BFS extracted feature sets (*p* < 0.05); ** shows statistical significance with *p* < 0.01; *** shows statistical significance with *p* < 0.001.

**Figure 7 bioengineering-10-00866-f007:**
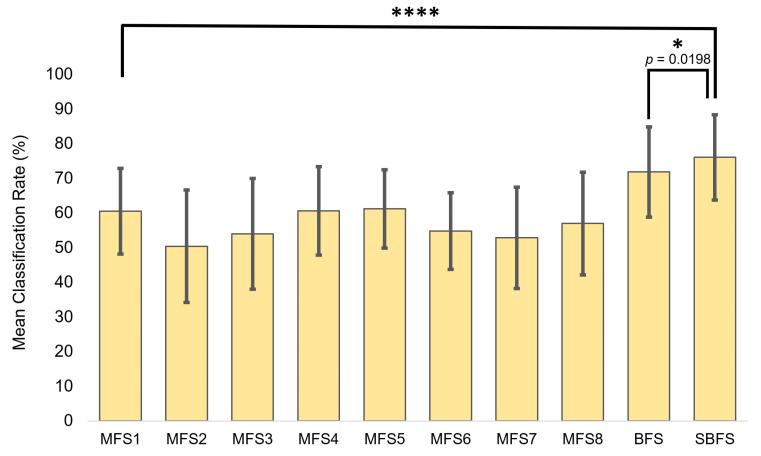
The superior feature sets obtained using brute-force searches (BFS) and semi-brute-force searches (SBFS), plus the use of MFSs canonical and multi-domain feature set comparisons to inspect the significance between classification rate (CCR) performance metrics. Asterisks indicate statistical significance in comparison to the mean classification rate using the unpaired *t*-test (* *p* < 0.05 and **** *p* < 0.0001).

**Figure 8 bioengineering-10-00866-f008:**
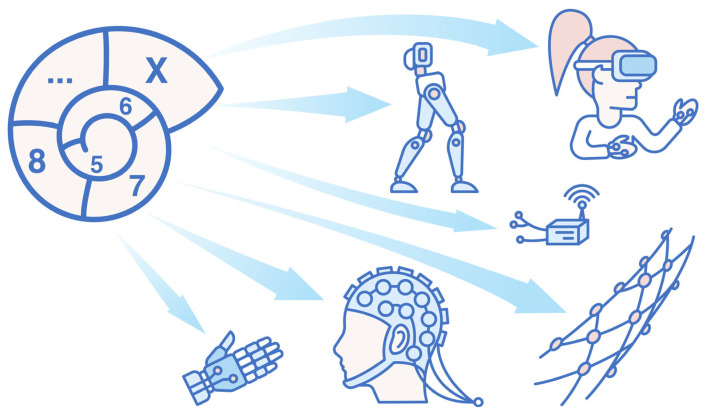
Semi-brute-force navigated amalgamation in the linkage of features for biosignal processing. The SNAiL algorithm is broadly applicable for numerous related fields in neurorehabilitation, such as lower-limb training via wearable exoskeletons, virtual and augmented realities (VR, AR), face paresis biofeedback electrical stimulations, cerebral palsy, spine injuries, remote diagnostics, precision medicine, patient-specific recovery monitoring, EMG-driven prosthesis control, intravascular neural interfaces, and EEG brain–computer interfaces.

**Table 1 bioengineering-10-00866-t001:** Post-acute stroke survivor characteristics (*n* = 19).

Participant Specifications	Data and Scores
Age, AV ± SD	65.9 ± 12.4
Gender, *n* (%)	
Male	12 (63.2)
Female	7 (36.8)
Race, *n* (%)	
Asian	19 (100)
Onset, days, AV ± SD	12.4 ± 6.3
Lesion characteristics, *n* (%)	
Cerebral infarction (CI)	11 (57.9)
Intracranial hemorrhage (ICH)	8 (42.1)
Affected side, *n* (%)	
Right	11 (57.9)
Left	8 (42.1)
Motor impairment scales	
FMA-UE (from 0 to 66) scores, AV ± SD	40 ± 20
SIAS-M (from 0 to 5), AV ± SD	[3 ± 1, 3 ± 2]
Brunnstrom stage (from 1 to 6), AV ± SD	[4 ± 1, 4 ± 1]
Hospitalization facilities, *n* (%)	
Stroke Care Unit (SCU)	14
Hospital ward	5

AV average value, SD standard deviation, FMA-UE Fugl-Meyer Assessment of the upper extremity after stroke, SIAS-M Stroke Impairment Assessment Set Motor (only the knee–mouth test and finger-function test). The Brunnstrom stage has binary parameters for the forearm and hand functional scoring.

**Table 2 bioengineering-10-00866-t002:** Extracted feature sets comparison in hemiplegic hand gesture prediction (*n* = 100 and equals the total number of 10-fold cross-validations).

Feature Set	CCR (%)	SD (%)	DF	*p*-Value
Best of SBFS	76.0714	12.3248	---	---
Best of BFS	71.8571	13.0339	198	0.0198 *
MFS1	60.5714	12.3710	198	4.1523 × 10^−16^ ****
MFS2	50.4286	16.2382	198	4.7378 × 10^−27^ ****
MFS3	54.0000	15.9661	198	4.2208 × 10^−22^ ****
MFS4	60.6429	12.7814	198	1.3719 × 10^−15^ ****
MFS5	61.2143	11.3169	198	4.0690 × 10^−16^ ****
MFS6	54.7857	11.0775	198	7.2799 × 10^−28^ ****
MFS7	52.8571	14.6068	198	9.8525 × 10^−26^ ****
MFS8	57.0000	14.8163	198	5.1484 × 10^−19^ ****

* shows statistical significance between the extracted feature sets on their classification capabilities (*p* < 0.05); **** shows significance with *p* < 0.0001. SBFS—semi-brute-force search, BFS—brute-force search, MFSs—multi-domain feature sets, SD—standard deviation, CCR—mean correct classification rate, and DF—degree of freedom during the unpaired *t*-test.

## Data Availability

In accordance with Japan’s data protection law, the Act on the Protection of Personal Information (APPI), and the decision of the Clinical Ethics Review Board, the data obtained from the research are not publicly accessible. For more information, please contact H.K.

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
