# Peer review of "Empirical Myoelectric Feature Extraction and Pattern Recognition in Hemiplegic Distal Movement Decoding"

_bioengineering, 2023, doi:10.3390/bioengineering10070866_

Round 1

Reviewer 1 Report

In this article, the authors introduce a method for acute-stroke gesture recognition through surface electromyography. This method is to investigate the brute-force analysis of individual and combinational feature vectors. Moreover, this study and combine post-brute force singular vectors and Fibonacci-like spiral ranking net for feature selection. Some comments are listed below.

1. In line 61 and 65, the statements “degrades the algorithm’s learning performance” and “negatively affect classification” are unclear, because there are many metrics to evaluate learning performance and classification level.

2. In line 119, it is necessary for authors to introduce the Fibonacci algorithm and its advantages over other algorithms in this part, in order to demonstrate the reasons for combining Fibonacci in this study.

3. The illustration of Figure 5 could be supplemented, such as what author want to demonstrate with Figure 5 and what conclusions to draw from Figure 5.

4. In line 375 and 401, why authors choose those features as these experimental sample? Whether it is completely random?

5. In line 569, whether the time cost is a limitation of this study? Because in line 407, this article mentions that “excluding pre-calibration, the interval calculation time took 1800 hours”.

None

Reviewer 2 Report

This study investigates brute-force (assumption-free) gesture recognition, using individual and combinational feature vectors from surface electromyography (EMG) of 19 patients, including concatenation of post-brute force singular vectors. The final semi-brute-force-search feature set achieved a 10-fold cross-validation classification rate of over 76%, about 10% higher than the various multi-domain feature sets. 

1. In Section 2.4, the brute-force feature selection search was stated to involve all combinations of 1, 2, 3 and 4 features, and the BFS4 feature set achieved the best CCR of about 71% (Figure 4). It might be clarified as to which sets of k features produced the best CCR, for each of BFS1 to BFS4.

2. In Section 2.4, it is stated that testing and training subsets were set to 17 and 2, but also that 10-fold cross-validation was performed. It might first be confirmed whether the proportion should be 17 training and 2 testing instead. It might be clarified as to whether the cross-validation was performed only on the training subset before applied to the test subset, or if the 10-fold cross-validation was done on all 19 participants, how the extra training subset was obtained (since 10 folds*2 participants = 20)

3. In Section 2.5, it is stated that BFS4 had 10,334,625 (i.e. over 10 million) combinations. It might be clarified as to the computational resources required to exhaustively perform cross-validation for all these combinations (1,800 hours briefly mentioned in Line 437).

4. In Section 2.6, eight multi-domain feature sets (MFS) are described, with a maximum of 14 features (MFS7). The proposed SNAiL in contrast includes up to 20 "feature-length generic subsets", which appears to simply mean "5 to 20 features" in Line 365. In this sense, it is probably not unexpected that the ability to include more (up to 20) features, would allow CCR to outperform compared to BFS (up to 4 features only) or MFS (up to 14 features only).

However, a natural question would then be whether it would be possible to just use all 127 available features to train a (SVM, or possibly LIBLINEAR) classifier, and whether that would further improve CCR. Moreover, there are standard methods for feature importance ranking & selection (to mitigate the possible overfitting mentioned in Line 506) especially with random forest/boosting based classifiers, which might be strongly considered.

5. In Section 4, "spiral concatenation" by SNAiL is mentioned (Line 531) for the first time; this might be explained in detail in the earlier methodology, assuming that it means something other than merely concatenating the selected features.

Round 2

Reviewer 1 Report

After the revision, the article can be accepted.

None

Reviewer 2 Report

We thank the authors for addressing our previous comments.